# The Association between *Tannerella forsythia* and the Onset of Fever in Older Nursing Home Residents: A Prospective Cohort Study

**DOI:** 10.3390/ijerph19084734

**Published:** 2022-04-14

**Authors:** Ayaka Koga, Wataru Ariyoshi, Kaoru Kobayashi, Maya Izumi, Ayaka Isobe, Sumio Akifusa, Tatsuji Nishihara

**Affiliations:** 1Division of Infections and Molecular Biology, Faculty of Dentistry, Kyushu Dental University, Fukuoka 803-8580, Japan; r20koga@fa.kyu-dent.ac.jp (A.K.); arikichi@kyu-dent.ac.jp (W.A.); r20kobayashi@fa.kyu-dent.ac.jp (K.K.); tatsujin@kyu-dent.ac.jp (T.N.); 2School of Oral Health Sciences, Faculty of Dentistry, Kyushu Dental University, Fukuoka 803-8580, Japan; r15izumi@fa.kyu-dent.ac.jp (M.I.); r17isobe@fa.kyu-dent.ac.jp (A.I.); 3ADTEC Co., Oita 879-0453, Japan

**Keywords:** fever, nursing home, older adults, periodontal pathogens, *Tannerella forsythia*

## Abstract

Background: Periodontal pathogens are related to the incidence of systemic diseases. This study aimed to examine whether periodontal pathogen burden is associated with the risk of fever onset in older adults. Methods: Older adults in nursing homes, aged ≥65 years, were enrolled. The study was set in Kitakyushu, Japan. The body temperatures of participants were ≥37.2 °C and were recorded for eight months. As periodontal pathogens, *Porphyromonas gingivalis*, *Treponema denticola*, and *Tannerella forsythia* were qualified by a real-time polymerase chain reaction at the baseline. For statistical analysis, the number of bacterial counts was logarithmically conversed to 10 as a base. Results: Data from 56 participants with a median age of 88 (62–98) years were available for analysis. The logarithmic-conversed bacterial counts of *T. forsythia*, but not *P. gingivalis* or *T. denticola*, were associated with the onset of fever in older residents. The Kaplan–Meier method revealed that the group with <10^4^ of *T. forsythia* had significantly less cumulative fever incidence than the group with ≥10^4^ of *T. forsythia*. The group with ≥10^4^ of *T. forsythia* was associated with an increased risk of fever onset (hazard ratio, 3.7; 98% confidence interval, 1.3–10.2; *p* = 0.012), which was adjusted for possible confounders. Conclusions: Bacterial burden of *T. forsythia* in the oral cavity was associated with the risk of the onset of fever in older nursing homes residents.

## 1. Introduction

Frail older adults tend to have a lower-than-normal temperature compared to younger adults; therefore, they are at risk of being nonfebrile [1]. Considering that older adults have lower-grade fevers compared with younger adults [2], it is particularly important not to overlook any symptom of a blunted fever in older adults. The most common causes of fever in older adults are respiratory infections [3]. Older adults have a high risk of developing aspiration pneumonia [4,5], a major etiology of morbidity and mortality in this population [6]. Respiratory tract infections in older adults often develop with a blunted fever [7] and without a cough or an increase in sputum production [8]. These features make it difficult to detect the infection and can result in exacerbation and a higher risk of mortality.

Recent studies have demonstrated that the microbiota in the oral cavity directly contribute to systemic inflammation through the hematogenous spread of bacterial toxins [9]. Bacteremia due to periodontal pathogens induce multipotent progenitors such as myeloid potential and granulocyte-macrophage progenitors, which promotes inflammation in other organs [10]. Oral inflammation has a deteriorative effect on several organ systems and increases the risk of developing systemic diseases, such as diabetes [11,12,13], cardiovascular diseases [14], low-weight birth [15], atherosclerosis [16], amyotrophic lateral sclerosis [17], and respiratory tract infection [18]. A previous case-control study demonstrated that people suffering from periodontitis had a three times higher risk for the development of hospital-acquired pneumonia compared with inpatients without periodontitis [19]. Several periodontal pathogens have been related to respiratory tract infections, including *Aggregatibacter actinomycetemcomitans*, *Capnocytophaga* spp., *Fusobacterium nucleatum*, *Fusobacterium necrophorum*, *Porphyromonas gingivalis*, and *Prevotella intermedia* [9]. Our previous study demonstrated that there is a significant association between oral trypsin-like activity and the risk of fever onset in older nursing home residents [20]. Although oral trypsin-like activity is considered to relate to the bacterial burden of *P*. *gingivalis*, *Treponema denticola*, and *Tannerella forsythia*, clinically designated red-complex periodontal pathogens [21,22], the association between the onset of fever and these pathogens remains unknown. We hypothesized that the bacterial burden of the red complex might be associated with the onset of fever in older adults. Thus, this study aimed to investigate whether the bacterial burden of *P*. *gingivalis*, *T. denticola*, and *T. forsythia* in the mouth is associated with the risk of the onset of fever in older nursing home residents.

## 2. Materials and Methods

### 2.1. Study Setting and Population

This prospective cohort study was conducted in ten nursing homes in Kitakyushu City, Fukuoka Prefecture, Japan, from February 2019 to December 2019. The participants of the study followed up for 8 months. This study was approved by the Kyushu Dental University Institutional Review Board for Clinical Research (No. 18–51). After explaining the purpose and procedure of the study, informed consent was obtained from all participants, their surrogates, or legal representatives.

### 2.2. Data Collection

Number of present teeth, periodontal pocket depths (PPDs), bleeding on probing (BOP), number of teeth, and comorbidity conditions at the baseline were collected. The axillary temperature was noted daily at 7:00 a.m. by the nursing home care staff. The repeated onsets of fever were recorded during the observation period. Physical health status details were obtained from the medical records maintained by the nursing homes. All data, excluding body temperature, were collected from 3:00 p.m. to 5:00 p.m. at the respective nursing homes at the baseline. An experienced dentist (S.A.) assessed PPDs, BOP, and the number of teeth. PPDs were assessed at six points (mesial, mid-, and distal points of the buccal and lingual sites). Comorbidity conditions were assessed using the Charlson comorbidity index [23,24].

### 2.3. Real-Time Polymerase Chain Reaction

Considering that the major cause of the onset of fever in older adults is aspiration, a reservoir of microbiota in the oral cavity, which includes the dorsum of the tongue, was selected to assess the bacterial burden of periodontal pathogens in the whole mouth. To collect the sample, the dorsum of the tongue of participants was swabbed using a cotton swab, with 10 strokes. After samples were suspended in a solvent, purification of the DNA derived from 100 μL of each suspended sample was performed using SMITEST EX-R&D (G&G Science Co., Ltd., Fukushima, Japan) according to the manufacturer’s protocol. For the detection of periodontopathic bacteria by polymerase chain reaction (PCR), the PCR products were detected using Brilliant III Ultra-Fast SYBR^®^ Green QPCR Master Mix with Low ROX (Agilent Technologies, Santa Clara, CA, USA) with the universal primers for the 16S rRNA gene and specific primer sequences listed in Table 1. Thermal cycling and fluorescence detection were performed using the AriaMx Real-Time PCR System (Agilent Technologies, CA, USA) with the PCR steps as follows: initial denaturation for 30 s at 95 °C, 50 cycles of 10 s at 95 °C for denaturation, 20 s at 60 °C for primer annealing, and 20 s at 72 °C for elongation. The quantity of each bacterium was determined with the previously established standard curves using homologous reference.

### 2.4. Outcome

The primary outcome of this study was the onset of fever during a follow-up period in the older nursing home residents.

### 2.5. Statistical Analyses

Descriptive statistics were used to characterize the participants. Each value was represented as median (minimum–maximum). The Mann–Whitney *U* test was used for continuous variables. Correlation between two variables was evaluated by Spearman’s rank correlation coefficient. For data analysis, bacterial counts were logarithmically conversed to 10 as a base. The participants were divided into two groups based on the results from the comparison of the bacterial burden between participants with or without the onset of fever during the observation period. These two groups were participants with <10^4^ and ≥10^4^ bacterial counts of *T. forsythia*. The incidence curves of initial fever onset during the eight months for participants in the two groups were analyzed using the Kaplan–Meier method. The Cox proportional hazards model was adopted to calculate hazard ratios (HRs) for the onset of fever in the participants. Sex, age, and comorbidity were identified as possible confounders of the association between logarithmic-conversed bacterial counts and the onset of fever. All analyses were performed using the International Business Machines (IBM) Statistical Package for the Social Sciences Statistics for Windows (IBM Corp., released 2012, version 22.0, Armonk, NY, USA). The Strengthening the Reporting of Observational studies in Epidemiology guidelines were followed while reporting the analysis of the observational data. The alpha error was set at 5%.

## 3. Results

For statistical analysis, data were obtained from 56 participants (20 men and 36 women). A total of 87 older adults aged ≥ 65 years were recruited. Ten participants refused to participate in the baseline survey, and 54 participants refused to undergo the collection of samples for RT-PCR. Nineteen participants left the nursing home before the onset of fever during the follow-up period, and two participants had died. Finally, 56 residents were included in this study. Figure 1 shows the process of participant selection through a flow diagram illustration. The median age of the participants was 88 (62–98) years. None of the participants were a current smoker. Moreover, 24 (42.9%), 31 (55.4%), and 15 (26.8%) participants had BOP, ≥4 mm PPD, and ≥6 mm PPD, respectively. Furthermore, 48 (85.7%), 38 (67.9%), and 55 (98.2%) participants had *P. gingivalis*, *T. denticola*, and *T. forsythia* pathogens, respectively.

The average time of the initial onset of fever from the baseline was 5.0 (1–12) days. Table 2 presents the comparison of bacterial counts of periodontal pathogens with the onset of fever. The Tf-log in the group with the onset of fever was significantly higher than that in the group without the onset of fever (4.2 (1.9–5.3) vs. 3.5 (0–5.6), *p* = 0.044), but not Pg-log or Td-log.

Taking these results into consideration, hereafter, statistical analysis was performed to compare the two groups (i.e., one group with Tf-log < 4 and another group with Tf-log ≥ 4). The baseline characteristics of the participants of this study are shown in Table 3. The group with Tf-log < 4 had significantly longer days until the first onset of fever than the group with Tf-log ≥ 4 (12 (1–12) vs. 2 (1–12), *p* = 0.027). The number of teeth with BOP, %BOP, and with ≥ 6 PPD were statistically higher in the group with Tf-log ≥ 4 than those in the group with Tf-log < 4 (0 (0–10) vs. 1 (0–10), *p* = 0.014; 0 (0–100) vs. 11.1 (0–63.0), *p* = 0.010; 0 (0–1) vs. 0 (0–12), *p* = 0.044, respectively). Age, sex, number of onsets of fever, comorbidity, body mass index, the number of teeth, and the number of teeth with ≥4 PPD did not indicate a statistical difference between the two groups. After excluding edentulous participants, number of teeth with BOP (0 (0–10) vs. 1 (0–10), *p* = 0.007) %BOP (0 (0–100) vs. 11.1 (0–63.0), *p* = 0.010), and 6 mm ≥ PPD (0 (0–1) vs. 1 (0–12), *p* = 0.026), but not 4 mm ≥ PPD (2 (0–16) vs. 5 (0–20), *p* = 0.065), also differed significantly between the two groups. In addition, there was no statistical difference in logarithmic-conversed bacterial counts of *P. gingivalis* (Pg-log), *T. denticola* (Td-log), and total bacteria between the two groups.

Next, we examined whether the bacterial burden of *T. forsythia* was associated with the duration of initial fever from the baseline. Figure 2 shows the empirical incidence curves of initial fever during eight months for the two groups based on the bacterial burden of *T. forsythia*. The Kaplan–Meier analysis revealed that the average dates until the onset of fever in the groups with Tf-log < 4 and ≥ 4 were 8.0 ± 1.0 and 4.1 ± 0.9 months, respectively. There was a significant difference in the cumulative onset of fever between the groups with Tf-log < 4 and ≥ 4 (long-rank test; *p* = 0.013). The incidence rates of onset of fever in the groups with Tf-log < 4 and ≥ 4 were 40.0% and 81.0%, respectively.

In the Cox’s regression models, the HRs for initial fever in the group with Tf-log ≥ 4 were 2.4 (95% confidence interval, 2.4 (1.1–5.3); *p* = 0.030) in the crude model and 3.7 (95% confidence interval, 1.3–10.2; *p* = 0.012) in the adjusted model for age, sex, and comorbidity, compared to the group with Tf-log < 4 (Table 4). To assess the validity of sample size for this analysis, when the observation period was eight months (0.67 year), the incidence rates of the onset of fever in Tf-log < 4 and Tf-log ≥ 4 were 0.4 and 0.8, respectively, with power and alpha error values of 0.8 and 0.05, respectively. The sample size of each group was calculated as 25.4, suggesting that the sample size of this study was appropriate.

## 4. Discussion

This is the first study to report that the bacterial burden of *T. forsythia* is associated with the onset of fever in older adults, to the best of our knowledge. *T. forsythia* is an anaerobic gram-negative, with a long rod-like segmented structure, and it is a member of the *Cytophaga–Bacteroides* family [25]. *T. forsythia* is frequently isolated from various periodontal lesions, including gingivitis and chronic and aggressive periodontitis [26], and it is closely related to the advance of loss of clinical attachment [27,28,29]. Our findings suggest that the bacterial burden of *T. forsythia* was associated with the number of teeth with BOP, or those with ≥6 mm PPD also supports this evidence. Several virulence factors driven from *T. forsythia* were reported, including trypsin-like protease [30]; two types of sialidases, SiaH [31] and NanH [32]; Bacteroides surface protein (Bsp) A [33]; a-D-glucosidase [34]; N-acetyl-β-glucosaminidase [34]; hemagglutinin [35]; components of the bacterial S-layer [36]; methylglyoxal [37]; karilysin [38]; forsythia detaching factor [39]. These virulence factors are considered to relate to immunodeficiency via inducing necrotic or apoptotic death in immune cells and bacterial invasion via the destruction of intracellular binding or connective tissues, followed by the apical migration of junctional epithelium and alveolar bone loss [40]. Based on an animal model, mice orally infected with *T. forsythia* showed increased alveolar bone loss [41]. However, *T. forsythia* inoculation did not cause any alveolar bone loss in gnotobiotic germ-free mice [41], suggesting that the coinfection of *T. forsythia* with other bacteria may be required for its virulence. This feature that exhibits its virulence was also confirmed in a previous study, which reported that *T. forsythia* co-inoculated with *F. nucleatum* or *P. gingivalis* induced abscess formation in rabbits. However, the mono infection of *T. forsythia* did not [42]. These lines of evidence suggest that the onset of fever observed in our present study may not be induced by the mono infection of *T. forsythia*. In this study, the synergic effect of *T. forsythia* and the other two pathogens (i.e., *P. gingivalis* or *T. denticola*) on the onset of fever was not observed. In cases of respiratory tract infection-related pyrexia, *T. forsythia* may interact with another oral or pharyngeal microbe. Despite the isolation of many virulence factors driven from *T. forsythia*, the association between systemic diseases and these factors remains unknown. Further studies on the role of *T. forsythia* in oral microflora or interaction with other pathogens on the onset of fever in older adults are required in the future.

In this study, there was no association between the bacterial burden of three pathogens and periodontal parameters, except between Tf-log and the number of teeth with BOP or ≥6 mm PPD. These results are possibly attributed to the sample-collected region, which was the dorsum of the tongue. A recent clinical study demonstrated that *P. gingivalis*, *T. forsythia*, and *A. actinomycetemcomitans* were isolated to subglottic levels regardless of periodontal health status [43], partly supporting our results. Notwithstanding, our results suggested that active inflammation of the periodontal pocket assessed by BOP or the remaining deep periodontal pockets might be a risk factor of the increased bacterial burden of *T. forsythia* in the oral cavity.

As another mechanism of exhibiting the pathogenicity of *T. forsythia*, a recent in silico study suggested that BspA driven from *T. forsythia* possibly induced immune responses via cross-reacting with human proteins [44]. The study argued that *T. forsythia* BspA shared peptide motifs with host proteins capable of inducing a stroke. This evidence suggests that *T. forsythia* may be able to induce immune responses with adverse effects on a host by its peptide epitope, regardless of its direct virulence exhibition to host organs. A high level of bacterial burden of *T. forsythia* may lower the threshold of immune response induction, or it may induce low-grade inflammation, followed by an exaggerated immune response.

Participants in this study included edentulous persons. A previous study demonstrated that large quantities of *A. actinomycetemcomitans*, *P. gingivalis*, and *T. forsythia* were isolated in edentulous patients undergoing orotracheal intubation [45], suggesting that the oral environment presenting favorable conditions for anaerobe growth, such as that with a thick tongue coat, may accumulate those pathogens even in the oral cavity without a tooth.

In this study, the bacterial burden of *P. gingivalis* was not associated with the onset of fever. The results from an animal experiment suggested that *P. gingivalis* was a causative agent of aspiration pneumonia in a gingipain-dependent manner [46]. A recent study demonstrated that the gingipain-induced expression of the platelet-activating factor receptor, a receptor for pneumococcus, and pneumococcal adhesion on host epithelial cells, but not lipopolysaccharide or fimbriae [47]. We may need to investigate the gingipain activity in order to elucidate the association between *P. gingivalis* and the onset of fever. 

Our finding that the bacterial burdens of *P. gingivalis* and *T. denticola* were not associated with the onset of fever does not deny the hematogenous infection of these pathogens from periodontal lesions. Hematogenous-spread periodontal pathogens induce systemic diseases, including cardiovascular disease [48] and Buerger’s disease [49]. Although many clinical and experimental studies have reported an association between periodontitis and respiratory tract diseases, the cause of infection at the respiratory tract is the direct inflow of these pathogens by aspiration rather than by a hematogenous manner [50,51,52]. This study aimed to investigate the association between the bacterial burden of the red complex in the whole oral flora and the onset of fever. To assess the bacterial burden of these pathogens in periodontal lesions, the sample should be collected from the periodontal pockets. This strategy can be performed in other studies.

During the follow-up period, the dates when the body temperatures of participants were more than 37.2 °C [20,53] were noted. In older individuals, if fever is defined as a body temperature ≥ 37.2 °C, the sensitivity and specificity of detecting infections are 83% and 89%, respectively [53]. The Infectious Diseases Society of America recommends 37.2 °C as a threshold for the onset of fever in older nursing home residents [54]. Thus, a temperature of 37.2 °C and above was defined as the onset of fever in this study.

There are some limitations to this study. First, the size of the cohort was 56 participants. Although the sample size was appropriate, a larger sample size might be required to increase reliability. Second, the origin or cause of fever was not assessed, as almost all the fever periods lasted for one day. The causes of fever onset are unclear; however, considering that the most common cause of fever in patients in the rehabilitation ward was respiratory tract infections [55], respiratory infections may also be a major cause of fever in the participants of this study. Third, since the participants of this study were residents of nursing homes in Japan, the results might not be the same for patients from other regions. This factor should be taken into consideration before generalizing the results. In addition, when the participating nursing homes were selected, there might have been selection bias.

## 5. Conclusions

In this study, the association between the bacterial burden of red complex periodontal pathogens in the oral cavity and the risk of the onset of fever in older nursing home residents was evaluated. The older adults with fever during the observation period have a significantly higher bacterial burden of *T. forsythia* than those without fever. Survival analysis revealed that the bacterial burden of *T. forsythia* was associated with the risk of the onset of fever in older adults. Mechanisms of how *T. forsythia* induces the onset of fever in older adults require further study.

## Figures and Tables

**Figure 1 ijerph-19-04734-f001:**
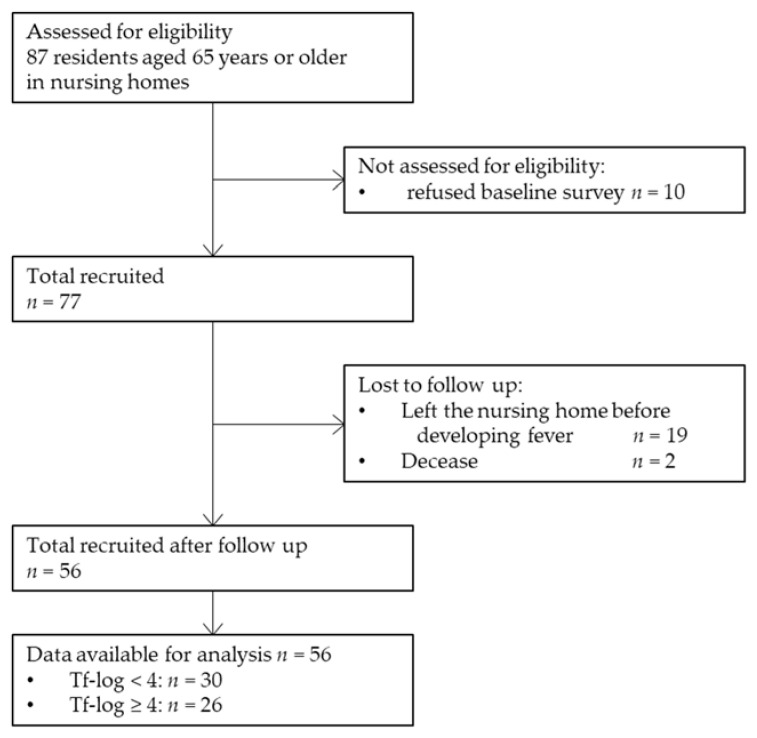
Flow diagram of the study participant selection. Tf-log: Logarithmic-conversed bacterial counts of *Tannerella forsythia*.

**Figure 2 ijerph-19-04734-f002:**
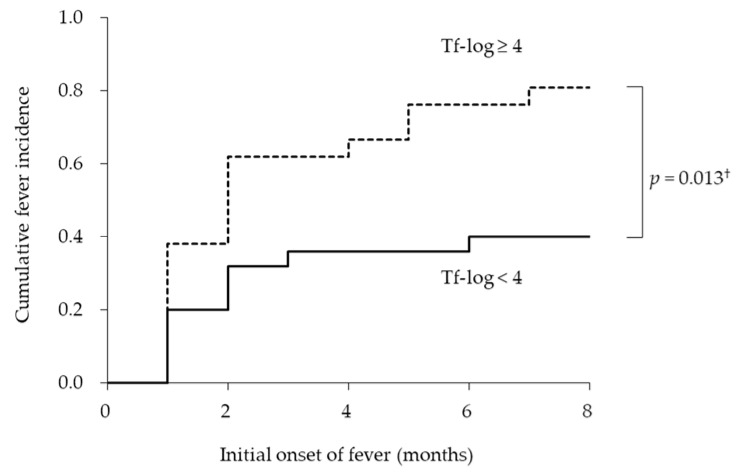
Incidence curves according to logarithmic-conversed bacterial counts of *Tannerella forsythia*. ^†^ log-rank test.

**Table 1 ijerph-19-04734-t001:** Primers for periodontal pathogens.

Bacteria	Primers
*Porphyromonas gingivalis*	Forward: CCGCATACACTTGTATTATTGCATGATATT
	Reverse: AAGAAGTTTACAATCCTTAGGACTGTCT
*Tannerella forsythia*	Forward: ATCCTGGCTCAGGATGAACG
	Reverse: TACGCATRCCCATCCGCAA
*Treponema denticola*	Forward: CCTTGAACAAAAACCGGAAA
	Reverse: GGGAAAAGCAGGAAGCATAA
Universal primer	Forward: TTAAACTCAAAGGAATTGACGG
	Reverse: CTCACGACACGAGCTGACGAC

**Table 2 ijerph-19-04734-t002:** Comparison of bacterial counts of periodontal pathogens with the onset of fever.

Bacterial Counts	Onset of Fever		*p*-Value ^†^
	(−) *n* = 28	(+) *n* = 28	
Pg-log	3.6 (0–6.4)	2.3 (0–5.6)	0.414
Td-log	0 (0–4.5)	0 (0–6.0)	0.060
Tf-log	3.5 (0–5.6)	4.2 (1.9–5.3)	0.044
Univ-log	7.1 (6.5–8.2)	7.2 (6.3–8.2)	0.610

^†^: Mann–Whitney *U*-test. Pg-log: logarithmic-conversed bacterial counts of *Porphyromonas gingivalis*; Td-log: logarithmic-conversed bacterial counts of *Treponema denticola*; Tf-log: logarithmic-conversed bacterial counts of *Tannerella forsythia*. Continuous values are described as median (minimum–maximum). Univ-log: logarithmic-conversed total bacterial count.

**Table 3 ijerph-19-04734-t003:** Characteristics of participants.

Variables	Tf-log < 4 (*n* = 30)	Tf-log ≥ 4 (*n* = 26)	*p*-Value
Age; *m*	90 (69–98)	86 (62–98)	0.175
Sex; *n* (%)			
Man	9 (30.0)	11 (42.3)	0.408
Woman	21 (70.0)	15 (57.7)	
Number of onsets of fever (days; *m*)	0 (0–17)	1 (0–16)	0.120
During initial onset of fever (days; *m*)	12 (1–12)	2 (1–12)	0.027
Charlson comorbidity index; *m*	1 (0–3)	1 (0–4)	0.773
Body mass index (kg/m^2^; *m*)	20.8 (16–24.9)	20.7 (14.5–27.3)	0.889
Number of teeth; *m*	5.5 (0–23)	11.5 (0-30)	0.411
BOP; *m*	0 (0–10) [0 (0–100)]	1 (0–10) [1 (0–10)]	0.014 [0.010]
%BOP; *m*	0 (0–100) [0 (0–10)]	11.1 (0–63.0) [11.1 (0–63.0)]	0.010 [0.007]
PPD ≥ 4 mm; *m*	0.5 (0–16) [2 (0–16)]	3.5 (0–20) [5 (0–20)]	0.160 [0.065]
PPD ≥ 6 mm; *m*	0 (0–1) [0 (0–1)]	0 (0–12) [1 (0–12)]	0.044 [0.026]
Pg-log; *m*	2.7 (0–5.8)	3.4 (0–6.4)	0.425
Td-log; *m*	0 (0–4.5)	0 (0–6)	0.642
Univ-log; *m*	7.1 (6.5–8.2)	7.2 (6.3–8.2)	0.402

Pg-log: logarithmic-conversed bacterial counts of *P. gingivalis*; Td-log: logarithmic-conversed bacterial counts of *T. denticola*; Tf-log: logarithmic-conversed bacterial counts of *T. forsythia*. Univ-log: logarithmic-conversed total bacterial count. BOP: bleeding on probing; PPD: periodontal pocket depth. %BOP was calculated as BOP divided by the number of teeth. The data in [ ] indicate results of analysis after excluding edentulous participants. Numbers of participants in groups Tf-log < 4 and Tf-log ≥ 4 were 22 and 19, respectively.

**Table 4 ijerph-19-04734-t004:** Cox’s hazard ratio for the onset of fever according to bacterial counts of *T. forthysia*.

	Crude Model			Adjusted Model ^†^		
Variable	B ± SE	HR (95% CI)	*p*-Value	B ± SE	HR (95% CI)	*p*-Value
Tf-log						
<4		1 (reference)			1 (reference)	
≥4	0.9 ± 0.4	2.4 (1.1–5.3)	0.030	1.3 ± 0.5	3.7 (1.3–10.2)	0.012

Tf-log: logarithmic-conversed bacterial counts of *Tannerella forsythia*. HR: hazard ratio; CI: confidential interval. ^†^: adjusted for sex, age, and Charlson comorbidity index.

## Data Availability

The data presented in this study are available upon request from the corresponding author. The data are not publicly available due to the inclusion of sensitive personal information.

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
