# Peer review of "The Association between Tannerella forsythia and the Onset of Fever in Older Nursing Home Residents: A Prospective Cohort Study"

_ijerph, 2022, doi:10.3390/ijerph19084734_

Round 1

Reviewer 1 Report

This manuscript reports on the relationship between oral bacteria and the onset of fever in aged persons at nursing homes. The authors concluded that T. forsythia other than P. gingivalis and T. denticola was associated with the risk of fever. 

The contents of this study are quite simple, however, there are some points should be improved as follows.

In general, the way of writing of M &M is poor, and extensive editing should be needed.

In "Data collection," the reason to decide the cut-off temperature should be described in Discussion not M &M

P. 2, Line 84: Since samples were obtained from the tongue dorsum, the description of "DNA was purified from saliva specimens" should be revised.

P. 4, L 147: What is "during the initial onset of fever"?

BOP should be expressed in percent.

Regarding the data of Table 1, the numbers of both total bacteria and P. g  were not significantly different between fever and non fever groups.  The authors should discuss this facts a bit.

The data of Table 2 were not be able to be understood. Explanation is completely insufficient.

Author Response

Replies to the comments of reviewer #1

This manuscript reports on the relationship between oral bacteria and the onset of fever in aged persons at nursing homes. The authors concluded that T. forsythia other than P. gingivalis and T. denticola was associated with the risk of fever.

The contents of this study are quite simple, however, there are some points should be improved as follows.

Reply: Thank you for your thoughtful review. The comments have helped significantly improve the quality of the manuscript, for which we are grateful.

In general, the way of writing of M &M is poor, and extensive editing should be needed.

In "Data collection," the reason to decide the cut-off temperature should be described in Discussion not M &M

Reply: Thank you for the helpful comment. Accordingly, the reason has been moved to the Discussion section.

2, Line 84: Since samples were obtained from the tongue dorsum, the description of "DNA was purified from saliva specimens" should be revised.

Reply: Thank you for your careful review. The revised sentence in line 86 has been highlighted in yellow.   

4, L 147: What is "during the initial onset of fever"?

Reply: Thank you for your comment. The sentence has been revised as follows, “until the first onset of fever” in line 148.

BOP should be expressed in percent.

Reply: Thank you for your helpful comment. The data for %BOP has been added in Table 2. The %BOP also shows a significant difference between the two groups.

Regarding the data of Table 1, the numbers of both total bacteria and P. g were not significantly different between fever and non fever groups.  The authors should discuss this facts a bit.

Reply: Thank you for your comment. As per the reviewer’s comment, the result in which P. gingivalis did not relate to the onset of fever was discussed in lines 239-245.

The data of Table 2 were not be able to be understood. Explanation is completely insufficient.

Reply: Thank you for your comment. In accordance with reviewer’s comment, explanations for the remaining variables have been added in lines 152-154.

Reviewer 2 Report

 Dear Authors,    

The study is of scientific interest and in line with the aims of the journal. The author guidelines have been respected and the work is well written.    

However, the manuscript should be improved and the manuscript should be copyedited by a native English speaker or copyediting service.    

Minor revisions: 

Abstract 

Please change “The body temperatures of participants were ≥37.2°C, and were were recorded for 8 months” to “ The body temperatures of participants were ≥37.2°C, and were recorded for 8 months”.   

Introduction

“Recent studies have demonstrated that the microbiota in the oral cavity directly contributes to systemic inflammation through the hematogenous spread of bacterial toxins”. Please explain the connection between periodontitis and systemic diseases, explaining how periodontitis can negatively affect the general state of health, especially in frail patients. Please cite:

- Ferrillo M et al. Periodontal Disease and Vitamin D Deficiency in Pregnant Women: Which Correlation with Preterm and Low-Weight Birth? J Clin Med. 2021 Oct 2;10(19):4578. doi: 10.3390/jcm10194578.

- Czerniuk MR, Surma S, Romańczyk M, Nowak JM, Wojtowicz A, Filipiak KJ. Unexpected Relationships: Periodontal Diseases: Atherosclerosis-Plaque Destabilization? From the Teeth to a Coronary Event. Biology (Basel). 2022 Feb 9;11(2):272. doi: 10.3390/biology11020272.

- de Sire A et al. Functional status and oral health in patients with amyotrophic lateral sclerosis: A cross-sectional study. NeuroRehabilitation. 2021;48(1):49-57. doi: 10.3233/NRE-201537.

- Merchant AT, Vidanapathirana N, Yi F, Celuch O, Zhong Z, Jin Q, Zhang J. Association between groups of immunoglobulin G antibodies against periodontal microorganisms and diabetes related mortality. J Periodontol. 2022 Feb 9. doi: 10.1002/JPER.21-0608.

The topic is very interesting and actual, and the work is well written.

In my opinion, the manuscript is suitable for publication in this Journal.

Author Response

Replies to the comments of reviewer #2

The study is of scientific interest and in line with the aims of the journal. The author guidelines have been respected and the work is well written.   

However, the manuscript should be improved and the manuscript should be copyedited by a native English speaker or copyediting service.   

Reply: Thank you for your serious-minded comments. Although the original manuscript was edited by a copyediting service, the revised manuscript has been re-edited by the copyediting service.

Minor revisions:

Abstract

Please change “The body temperatures of participants were ≥37.2°C, and were were recorded for 8 months” to “ The body temperatures of participants were ≥37.2°C, and were recorded for 8 months”.  

Reply: Thank you for your comment. The extra “were” has been deleted.

Introduction

“Recent studies have demonstrated that the microbiota in the oral cavity directly contributes to systemic inflammation through the hematogenous spread of bacterial toxins”. Please explain the connection between periodontitis and systemic diseases, explaining how periodontitis can negatively affect the general state of health, especially in frail patients. Please cite:

- Ferrillo M et al. Periodontal Disease and Vitamin D Deficiency in Pregnant Women: Which Correlation with Preterm and Low-Weight Birth? J Clin Med. 2021 Oct 2;10(19):4578. doi: 10.3390/jcm10194578.

- Czerniuk MR, Surma S, Romańczyk M, Nowak JM, Wojtowicz A, Filipiak KJ. Unexpected Relationships: Periodontal Diseases: Atherosclerosis-Plaque Destabilization? From the Teeth to a Coronary Event. Biology (Basel). 2022 Feb 9;11(2):272. doi: 10.3390/biology11020272.

- de Sire A et al. Functional status and oral health in patients with amyotrophic lateral sclerosis: A cross-sectional study. NeuroRehabilitation. 2021;48(1):49-57. doi: 10.3233/NRE-201537.

- Merchant AT, Vidanapathirana N, Yi F, Celuch O, Zhong Z, Jin Q, Zhang J. Association between groups of immunoglobulin G antibodies against periodontal microorganisms and diabetes related mortality. J Periodontol. 2022 Feb 9. doi: 10.1002/JPER.21-0608.

Reply: Thank you for your helpful comments. The explanation of the negative effect of periodontal pathogens on general health has been added in lines 41-43, written in red font. The references suggested by the reviewer have been added as #13, #15, #16, and #17.

Reviewer 3 Report

Dear Authors, 

The topic in interesting but manuscript has serious flaws in methodology:

  1. Abstract line 15  -delete one "were"
  2. Materials and Methods: why did You take the sample from the tongue? The method is not clear. You mention and made statistics with PPD and BOP and in Discussion line 230 You stated "Participants in this study were edentulous". How did You perform BOP and PPD if the patients were edentulous?  
  3. Did You record just first onset of fever or did You record repeating of fever onset?
  4. In Discussion part - lines 189-187 belong to Introduction, not discussion (several virulence...bone loss). 

Author Response

Replies to the comments of reviewer #3

The topic in interesting but manuscript has serious flaws in methodology:

Reply: Thank you for your thoughtful review. I hope that the revision reflects your intentions.

Abstract line 15  -delete one "were"

Reply: Thank you for your comment. The extra “were” has been deleted.

Materials and Methods: why did You take the sample from the tongue? The method is not clear. You mention and made statistics with PPD and BOP and in Discussion line 230 You stated "Participants in this study were edentulous". How did You perform BOP and PPD if the patients were edentulous? 

Reply: Thank you for your critical comments. Considering that the major cause of the onset of fever in older adults is aspiration, a reservoir of microbiota in the oral cavity, which includes the dorsum of the tongue, was selected to assess the bacterial burden of periodontal pathogens in the whole mouth. The reason has been stated in section 2.3. Real-Time Polymerase Chain Reaction, which is indicated in red font.

As the reviewer indicated, the statement, "Participants in this study were edentulous" was a mistake. I apologize for the confusion. We have corrected it as follows, “Participants in this study included edentulous persons”. 

Did You record just first onset of fever or did You record repeating of fever onset?

Reply: Thank you for your comment. The repeated onsets of fever were recorded. This has been added in section 2.2.

In Discussion part - lines 189-187 belong to Introduction, not discussion (several virulence...bone loss).

Reply: Thank you for your comment. The present study revealed the association between the onset of fever and T. forsythia. Thus, it seems natural to discuss the association between the onset of fever and the virulence factor of T. forsythia in the discussion section.

Round 2

Reviewer 1 Report

The manuscript has been revised properly.

Author Response

The manuscript has been revised properly.

Reply: Thank you for your variable review. The reviewer’s comments made a significant contribution to improve the quality of the manuscript.

Reviewer 3 Report

Dear Authors, 

you have clarified the part about toothless patients. 

Regarding BOP and PPD, You should make statistics only with those patients who had teeth. 

Author Response

Reviewer's comment

You have clarified the part about toothless patients.

Regarding BOP and PPD, You should make statistics only with those patients who had teeth.

Reply: We added: “After excluding edentulous participants, %BOP (0 [0–100] vs. 11.1 [0–63.0], p = 0.010), number of teeth with BOP (0 [0–10] vs. 1 [0–10], p = 0.007) and 6 mm PPD (0 [0–1] vs. 1 [0–12], p = 0.026), but not 4 mm PPD (2 [0–16] vs. 5 [0–20], p = 0.065), also differed significantly between the two groups.” The change is shown in red font in Lines 155–159.

Round 3

Reviewer 3 Report

Dear Authors, 

new data about PPD and BOP should also be changed in the Table 2. 

Author Response

Thank you for your comments. According the reviewer's comment, Table s was changed.